# Surface Engineering Methods for Powder Bed Printed Tablets to Optimize External Smoothness and Facilitate the Application of Different Coatings

**DOI:** 10.3390/pharmaceutics15092193

**Published:** 2023-08-24

**Authors:** Khanh T. T. Nguyen, Daan Zillen, Franca F. M. van Heijningen, Kjeld J. C. van Bommel, Renz J. van Ee, Henderik W. Frijlink, Wouter L. J. Hinrichs

**Affiliations:** 1Department of Pharmaceutical Technology and Biopharmacy, University of Groningen, 9700 RB Groningen, The Netherlands; t.t.k.nguyen@rug.nl (K.T.T.N.); d.zillen@rug.nl (D.Z.); h.w.frijlink@rug.nl (H.W.F.); 2The Netherlands Organization for Applied Scientific Research (TNO), 5656 AE Eindhoven, The Netherlands; franca.vanheijningen@tno.nl (F.F.M.v.H.); kjeld.vanbommel@tno.nl (K.J.C.v.B.); renz.vanee@tno.nl (R.J.v.E.)

**Keywords:** 3D printing, powder bed printing, personalized medicine, biologics, formulation, surface analysis, ileo-colonic targeting, film coating, ColoPulse, controlled release

## Abstract

In a previous attempt to achieve ileo-colonic targeting of bovine intestinal alkaline phosphatase (BIAP), we applied a pH-dependent coating, the ColoPulse coating, directly on powder bed printed (PBP) tablets. However, the high surface roughness necessitated an additional sub-coating layer [Nguyen, K. T. T., *Pharmaceutics* 2022]. In this study, we aimed to find a production method for PBP tablets containing BIAP that allows the direct application of coating systems. Alterations of the printing parameters, binder content, and printing layer height, when combined, were demonstrated to create visually less rough PBP tablets. The addition of ethanol vapor treatment further improved the surface’s smoothness significantly. These changes enabled the direct application of the ColoPulse, or enteric coating, without a sub-coating. In vitro release testing showed the desired ileo-colonic release or upper-intestinal release for ColoPulse or enteric-coated tablets, respectively. Tablets containing BIAP, encapsulated within an inulin glass, maintained a high enzymatic activity (over 95%) even after 2 months of storage at 2–8 °C. Importantly, the coating process did not affect the activity of BIAP. In this study, we demonstrate, for the first time, the successful production of PBP tablets with surfaces that are directly coatable with the ColoPulse coating while preserving the stability of the encapsulated biopharmaceutical, BIAP.

## 1. Introduction

Solid oral formulations are the predominant dosage forms for most pharmaceuticals on the market. As medical treatments become increasingly personalized, the preparation methods for solid dosage forms are also evolving. Over the last decade, several new oral dosage forms have gained increased attention, for example, tablets that contain APIs in a liquid state [1] and solid oral foams that increase gastroretention [2,3]. Another example is the application of additive manufacturing methods to produce new oral dosage forms. Originally created as a means to produce prototypes in the industry, the high level of customization has led three-dimensional printing (3DP) to be used in the pharmaceutical industry. Compared to traditional production methods, 3DP might result in reduced labor and resource investment and allow rapid prototyping, especially due to its inherent dose flexibility, as reviewed by Trenfield et al. [4]. Dosage forms that vary in structure, design, drug load, and drug release profile to fit different needs can be created through 3DP [5,6]. This versatility, together with the flexibility to manufacture in small volumes, can greatly benefit drug development as well as clinical testing of newly developed drugs that exist in limited quantities [5].

Powder-bed printing (PBP), also known as binder jetting, is a layer-by-layer manufacturing process. Initially, a fixed amount of powder is distributed to obtain an even layer of powder. The layer is then wetted at specific coordinates, corresponding to the desired design, by a binding fluid applied via an inkjet printhead. Then a new layer of powder is distributed on top of the initial layer. The previously (partially) dissolved powder will bind both the previous powder and the newly deposited powder together. This process is repeated for multiple layers until the final structure is produced [5,6,7]. A final drying step is required to remove residual solvent and fully solidify the layers, usually with convection ovens or IR light. Among the several techniques being studied for 3DP of drugs, PBP is the first and still the only technique that resulted in an approved drug product by the US Food and Drug Administration (FDA) in 2015, namely Spiritam^®^, a rapid orodispersible formulation containing levetiracetam [8]. When working with biopharmaceuticals, PBP might be more suitable than other often used 3DP methods, such as fused deposition modeling and laser sintering, as the high temperatures used in these methods might be detrimental to protein stability [9].

Coating is a widely employed method to achieve a controlled release of the drug, with many coating formulations having been produced and marketed. However, only a few studies have tried to apply different coating formulations to 3D-printed tablets to achieve different release profiles [9,10]. Tablets produced by PBP often have high surface roughness and porosity [6,11,12]. Most reported porosity values range between 50 and 60% [13,14,15] and, in certain cases, even up to 80% [16,17]. This can pose a major challenge when applying a coating to PBP tablets, as these coatings should be fully closed to function properly [9].

Previously, successful coating of PBP tablets was achieved only after the deposition of a sub-coating layer consisting of PEG 1500, which was required to obtain a smoother tablet surface [9]. Although this approach proved to be effective, it added an extra coating step and thus increased the complexity of the production process. Hence, a direct coating step is desirable.

The formulation design for PBP tablets has been studied extensively [11,17,18,19,20,21]. To circumvent the use of PEG 1500, it might be feasible to manipulate the formulation and printing process to achieve a sufficient degree of tablet surface smoothness to allow direct coating. Reducing layer thicknesses, also known as layer height, and increasing the binder content resulted in firmer, less porous, and thus possibly easier-to-coat tablets [11,15]. We also imagined that exposure to ethanol vapor might also help reduce surface roughness. Therefore, this study aimed to investigate the aforementioned parameters to allow the creation of tablets with surfaces that are directly coatable. Additionally, the effect of these changes as well as the subsequent coating process on the stability of the incorporated alkaline phosphatase, a possible biopharmaceutical for the treatment of ulcerative colitis, was also investigated.

## 2. Materials and Methods

### 2.1. Materials

Hydroxypropyl cellulose LFP (HPC-LFP) was obtained from Nisso HPC (Nisso Chemical Europe GmbH, Düsseldorf, Germany). Bovine intestinal alkaline phosphatase (BIAP), bovine serum albumin (BSA), ammediol (2-Amino-2-methyl-1,3-propanediol), HEPES (4-(2-hydroxyethyl)-1-piperazineethanesulfonic acid), and triethyl citrate were obtained from Sigma-Aldrich (St. Louis, MO, USA). D-mannitol was purchased from VWR Chemicals, BDH, USA. Eudragit S100 and Eudragit L100-55 were gifts from Evonik Operation GmbH (Essen, Germany). Macrogolum 6000 (PEG6000) and talc were obtained from BUFA (IJsselstein, The Netherlands). Croscarmellose sodium (AcDiSol) was obtained from FMC BioPolymer (Philadelphia, PA, USA). Inulin (4 kDa) was a generous gift from Sensus (Roosendaal, The Netherlands). Methylene blue was obtained from Interpharm B.V. (Rotterdam, The Netherlands; now discontinued).

### 2.2. Spray Drying of Inulin Stabilized BIAP

The HEPES buffer 2 mM pH 7.4 solution was heated to the boiling point before dissolving inulin 4 kDa and then left to cool to ambient temperature. BIAP was then added to the inulin solution at a 99:1 inulin:enzyme ratio (*w*/*w*) to create 5% (*w*/*v*) solutions [22]. A Büchi B-290 mini spray drier equipped with a high-performance cyclone, a B-296 dehumidifier, and a B-295 inert loop (Flawil, Switzerland) was used to spray dry the solution. Parameters for spray-drying were: 105 °C inlet temperature, 3.3 mL/min feed rate, 50 mm atomizing airflow, and 100% aspirator airflow. The spray-dried powder was collected, and the container was filled with dry nitrogen gas, sealed, and stored at 2–8 °C until it was used for printing.

### 2.3. Powder Mixture Preparation and PBP

The binder HPC-LFP and bulking agent mannitol were passed through a 100 µm sieve. The sieved powders were blended at 24 rpm for 30 min in a tubular mixer, a Stuart General Rotator STR4 (Reagecon, Shannon, Ireland), with an STR4/3 drum (Antylia Scientific, Vernon Hills, IL, USA), to produce the printing powder. For the preparation of tablets containing spray-dried (SD) inulin/BIAP, geometric dilution was carried out initially with SD inulin/BIAP and mannitol in a stainless-steel mortar. The remaining binder:bulk powder, and the mixed mannitol/SD inulin/BIAP were then added together into a tubular mixer for mixing at 24 rpm for 30 min.

Tablets were printed using an in-house-built powder bed printer developed by TNO [23]. The modeled 3D structures and the printing procedure were the same as previously described by Nguyen et al., as presented in Figure 1 [9]. In short, the tablet 3D models were designed with OpenSCAD (version 2019.05), exported to STL format files, and sliced using Simplify3D (version 4.0.1) to generate G-code instructions for the powder bed printer. The printing powder was transferred from a powder depositor onto the printing platform and spread out evenly to create a layer of consistent thickness using a counter-rotating roller. A solenoid valve (Lee valve INKA2436510H, orifice diameter of 70 μm, and FFKM seal material) with a drop mass of 11–12 μg was used for jetting ethanol selectively onto the printing powder as designed (Figure 1). The distance between two lines of the jetting liquid being deposited is called line spacing (LS). For tablets used in in vitro dissolution testing, methylene blue was dissolved in the jetting fluid to function as a release indicator. Tablets were dried overnight in an oven at 50 °C. Finally, the tablets were placed in a container, vacuum sealed, and stored at 2–8 °C until further analysis.

Tablets were treated with ethanol vapor as follows: Three-dimensionally printed tablets were placed on a sponge and then placed in a plastic box on top of an inverted aluminum dish. 50 mL of ethanol was poured onto the bottom of the box, after which the box was closed. After a certain time, depending on the experiment, the tablets were taken out and placed in an oven at 50 °C for an hour to dry. Tablets were taken out and stored at 2–8 °C in an atmosphere of dry nitrogen until further analysis.

### 2.4. Basic Tablet Properties Analysis

Tablets were analyzed for physical properties including diameter, thickness, weight, crushing strength, friability, and disintegration time. Tablets’ diameters and thicknesses were measured with a Digital ABS AOS Caliper (Mitutoyo, Veenendaal, The Netherlands). Tablets were weighed using an AE200 Analytical Balance (Mettler Toledo, OH, USA). Tablets’ crushing strengths were evaluated using the Pharmatron 6D tablet hardness tester (Dr. Schleuniger, Solothurn, Switzerland). Due to the inherent porosity of PBP tablets, some of them did not exhibit cracking during the crushing strength experiment; thus, the device would keep running and flatten the tablet. Results from these incidents were excluded from further analysis. Friability was determined with a single-blade friability tester, Erweka (Erweka, Hessen, Germany). In total 10 tablets were dedusted, weighed, and placed carefully in the rotor, which rotated at 20 rpm for 5 min. Afterward, tablets were dedusted and weighed again to calculate the relative weight loss. The disintegration times of tablets were determined using a European Pharmacopoeia 7.0 standard DT2 disintegration tester (Sotax, Aesch, Switzerland). Demineralized water at 37 °C was used as the disintegration medium. All measurements were performed in triplicate, except weight determination, which was conducted for 10 tablets.

### 2.5. SEM Surface Imaging of Printed Tablets

Tablet surface morphology was analyzed. Surface morphology was visualized using a scanning electron microscope (SEM) JSM 6460 (JEOL, Tokyo, Japan) as described previously [24]. Tablets were fixed on sample stubs by a double-sided adhesive carbon tape, sputter coated with 10 nm of pure gold using a JFC-1300 auto fine coater (JEOL, Tokyo, Japan), and purged in argon gas before being put under a high vacuum. Imaging was carried out with a spot size of 25, an acceleration voltage of 10 kV, and a working distance of 10 mm.

### 2.6. Application of the ColoPulse and Enteric Coating

The ColoPulse coating consisted of Eudragit S100:PEG6000:AcDiSol:Talc in a weight ratio of 7:1:3:2 dissolved or dispersed in 96% ethanol [25]. The enteric coating contained Eudragit L100-55:triethylcitrate:Talc 10:1:2.85 (*w*/*w*/*w*) and was also dissolved or dispersed in ethanol (96%). The coating setup consisted of a mini-rotating drum at 32 rpm equipped with a nozzle with a bore diameter of 1 mm (Schlick 970, Düsen-Schlick, Coburg, Germany) driven by a connected peristaltic pump (Minipuls 3, Gilson, Viliers le Bel, France) at a spray rate of 0.75–1.0 mL/min. Dummy tablets made of similar dimensions, each weighing 150 mg, were added together with the PBP tablets during coating to assist the tablet’s tumbling and ensure proper distribution of the coating being sprayed. The total number of PBP tablets and dummies was 40 for every coating batch. The drum temperature was maintained within 20–25 °C. After coating, the tablets were left to dry in the rotating drum for 5 min before being transferred into an oven for drying at 30 °C for 2 h.

### 2.7. Assessment of BIAP Stability

BIAP stability was determined by measuring its enzymatic activity [22,26]. A colorless substrate, p-nitrophenyl phosphate (p-NPP), which turns yellow after enzymatic conversion and can be detected at 415 nm, was used. Tablets were cut in half and dissolved in ultrapure water containing 0.01% BSA (*w/w*) to 10 µg/mL BIAP, based on total tablet weight. Then, 20 µL of sample solution was added to 160 µL of 0.05 M ammediol (pH 9.8) containing 1% MgCl_2_ (*w*/*v*) in a 96-well plate. Mixtures were equilibrated on a stove at 37 °C for 10 min. 20 µL of 5 mg/mL p-NPP was subsequently added to start the reaction, and the color development was measured at 415 nm every 30 s for 5 min using a Synergy HT plate reader (BioTek, VT, USA) equilibrated at 37 °C. In between each measuring interval, the microplate was shaken for 15 s. The activity of samples was calculated based on the conversion rate of p-NPP, which was fitted to the calibration curves of the reference BIAP solutions ranging from 0 to 10 µg/mL using linear regression analysis.

### 2.8. Drug Release Profile Testing in the Gastrointestinal Simulated System

The in vitro dissolution of the coated tablets was evaluated in a gastrointestinal simulation system (GISS) previously described by Schellekens et al., with the modification that half of the volumes were used [27]. In GISS, tablets are exposed to four different phases of different pHs, similar to the passage in the in vivo gastrointestinal tract. These phases are shown in Table 1. The GISS was prepared by adding four different media sequentially, as presented in Table 2. Approximately 5–10 min before reaching the next phase time point, the subsequent media were added using a peristaltic pump. The composition of each medium is provided in Table 2. Tablet dissolution testing was carried out in a USP dissolution apparatus type 2 (Sotax AT 7, Sotax, Basel, Switzerland) at 37 °C and a paddle speed of 50 rpm. The release of tablet content was determined by measuring the concentration of methylene blue using an in-line UV-spectrophotometer set to measure at 664.5 nm (Evolution 300 UV–VIS spectrophotometer, Thermo Fisher Scientific, Madison, WI, USA). Samples were taken every 5 min for 8 h, and all experiments were performed in triplicate.

### 2.9. Statistical Analysis

Prism 8.0 was used for data analysis. Data are presented as means ± standard deviation and analyzed using one-way ANOVA with post hoc Tukey’s multiple comparison test. Results with a calculated *p* < 0.05 were considered significantly different.

## 3. Results and Discussions

### 3.1. Influencing Tablet Surface Characteristics

#### 3.1.1. The Effect of Printing Conditions

The primary focus was on obtaining a more homogeneous and smoother surface for PBP tablets. Some formulation factors have demonstrated positive effects on reducing the porosity of the tablets and, as a result, the surface roughness. They include the particle size of materials, the layer thickness during printing, and the binder content [11,15]. The PBP tablets developed in our previous study had a rough surface and a layered ‘staircase’ structure on the tablet’s side, resulting in incomplete coating coverage, especially due to the irregular tablet’s side [9]. The coating droplets were most likely not big enough to cover the gap between each layer. Reducing the surface roughness and minimizing this staircase gap were therefore vital. This was evaluated by modifying the composition of the formulation and the printing conditions. The tablets printed for these evaluations did not include any SD inulin/BIAP powder. Five different types of tablets varying in binder:bulk ratio, line spacing, and layer height were prepared (see Table 3). Tablets formulated according to our previous study were used as a starting point for comparison (tablet A) [9]. The following modifications were applied: The binder:bulk ratio was increased from 20:80 to 50:50 (tablet B), the line spacing was increased from 0.45 to 0.50 mm (tablet B to C), and the layer height was reduced from 0.4 to 0.2 (C to D) and 0.1 (C to E). The small increase in line spacing from tablet B to C was made to accommodate the subsequent gradual decrease in layer height. Keeping the 0.45 mm line spacing would lead to the overexposure of ethanol at each printing layer at a lower height.

The tablet surface was evaluated by analyzing SEM micrographs (Figure 2). An increase in the binder content (from tablet A to tablet B), while showing a visually improved surface smoothness, did not yield a similar beneficial effect on the smoothness of the tablet’s side, specifically the staircase structure. Therefore, the layer height was gradually reduced from tablets C 0.4 mm (tablet A–C) progressing to 0.2 mm (tablet D), and finally 0.1 mm (tablet E). SEM micrographs revealed a notable reduction in the staircase structure, as it nearly disappeared. The small increase in line spacing from tablet B to C did not have much effect on the tablet. Tablets D and E visually appeared to have a more compact outer structure, and the tablet sides showed decreasing layer steps as the layer thickness was reduced from 0.4 to 0.2 and 0.2 to 0.1 mm, Figure 2C–E, respectively. Nevertheless, the loosely connected particles, an inherent characteristic of PBP tablets, remained, presenting an inhomogeneous surface with lots of small gaps in between.

#### 3.1.2. The Effect of Ethanol Vapor Treatment

To further increase surface smoothness, the printed tablets were exposed to ethanol vapor. It was hypothesized that during this process, the binder on the tablet’s surface would partially dissolve, leading to the filling of pores by viscous flow, followed by re-solidification after drying. All tablets A, B, C, D, and E were subjected to the ethanol vapor treatment process, which lasted for 100 min. Afterward, the surface of the tablets was analyzed by SEM. Following ethanol vapor treatment, all tablet types A to E showed an improvement in their visual roughness (Figure 3). Tablets A, B, and C did show changes in surface characteristics after exposure to ethanol; however, many pores could still be observed. Tablet A showed the least noticeable change among all the tested tablets. Upon subsequent testing, extended treatment time could not improve the surface smoothness of tablet A further. They might have lower susceptibility to ethanol vapor treatment due to the lower binder content. The partially dissolved binder on the tablet A surface was probably insufficient to fill the gaps between the loosely connected materials during ethanol vapor exposure. On the other hand, tablets D and E show the most drastic changes in surface profiles observed by SEM, with most of the original roughness due to loosely connected particles and especially the layered structure reduced to a smooth surface. Visually, tablets D and E appeared to be similar in terms of surface characteristics.

#### 3.1.3. The Influence of Surface Manipulation Processes on the General Properties of PBP Tablets

As the smoothness of the tablet’s surface was improved by these changes in the formulation and/or printing parameters, it is important to consider their effects on the tablet’s physical properties. Therefore, the weight, dimension, friability, crushing strength, and disintegration time of the different tablets before and after ethanol vapor treatment were determined (results see Figure 4). The tablet weight was, as expected, not significantly affected by ethanol vapor treatment for all tablets. Tablets D and E exhibited significantly higher weight than other tablets, while the tablet dimensions were similar, suggesting a denser structure and thus lower porosity. This can be attributed to the lower layer thickness, which resulted in greater jetting liquid (ethanol) exposure and less powder being printed per layer. As a consequence, the binding of particles increased, resulting in a lower porosity and thus an increased tablet weight. Tablet height slightly decreased after ethanol vapor exposure; however, this decrease was neither significant nor consistent across the various tablet types.

Friability data showed the most drastic changes between tablets before and after ethanol vapor treatment. The friability of tablets A, B, and C was 1.8 to 3% before ethanol vapor treatment but was reduced to well below 1% after ethanol vapor treatment. Tablets D and E, however, already had less than 1% friability, even without ethanol treatment. Crushing strength and disintegration time increase with high binder content (tablet A to B) and lower layer thickness (tablet C to D, and E). Both crushing strength and disintegration time of the tablets did not show significant differences before and after ethanol vapor treatment, except for tablet C. Despite having the same binder content as tablet B and higher than tablet A, tablet C had much higher friability, disintegrated faster, and showed lower crushing strength than tablets A and B. This was possibly due to the higher line spacing, as the reduced ethanol exposure led to insufficient particle connection from the binder. As a result, tablet C was much more fragile, more prone to cracking in the crushing strength test, and disintegrated faster. The addition of ethanol vapor treatment could thus show a significant improvement in tablet C’s friability, crushing strength, and disintegration time. Tablets D and E had noticeably higher crushing strengths and disintegration times (approximately 60–70 min) than tablets A, B, and C, regardless of an ethanol vapor treatment. The higher tablet weight, lower friability, higher crushing strength, and longer disintegration time point toward a well-connected structure, which was also seen in the SEM images (Figure 2 and Figure 3). These changes can be attributed to two factors: the lower layer thickness, leading to increased binding caused by the jetting liquid, and, more importantly, the higher binder content in the printing powder, as also noted in other studies [28,29].

#### 3.1.4. Optimization of Ethanol Vapor Treatment Time for Tablet E

Of all tablet types investigated, tablet E visually had the least rough surface and smoothest side, especially after ethanol vapor treatment. Because tablet E also had excellent crushing strength and friability, it was selected for subsequent experiments on coating and stability evaluation of incorporated inulin-encapsulated BIAP. An optimization of vapor treatment was conducted to reduce ethanol exposure to BIAP and facilitate the production process. Tablets were exposed to ethanol vapor over different time intervals and then visualized by SEM to observe this process. With increasing ethanol vapor exposure time, the surface of the tablets appeared increasingly smoother (Figure 5). After 60 min of treatment, a complete filling of the gaps was observed (Figure 5D). Thus, it was concluded that a treatment time of 60 min was sufficient to manipulate PBP tablet E surface roughness toward a smoother and more continuous structure.

### 3.2. Coating, SEM Imaging, and In Vitro Release Profile of the Coated 3D Printed Tablets

Tablets E, without or with ethanol vapor treatment (60 min), were coated with either a ColoPulse or an enteric coating layer without first applying a PEG sub-coating. To examine whether the coated tablets exhibited the desired release profile, in vitro drug release was tested for the coated tablets using the GISS.

For the ColoPulse coating, three different coating thicknesses of 8, 10, and 12 mg/cm^2^ were applied to the tablets. In Figure 6, the release profiles of tablet E with and without ethanol vapor treatment are given. Both tablets released the methylene blue only after the dissolution medium was changed to phase III, simulating the terminal ileum, which conformed with the desired illeocolonic targeting profile. The release rate decreased with increasing thicknesses of the ColoPulse coating. Whereas a coating thickness of 8 mg/cm^2^ resulted in an almost complete release after the last 4 h, tablets coated with 10 and 12 mg/cm^2^ did not reach 100% cumulative release. However, when measuring the methylene blue concentration in the dissolution medium after an additional 4 h, up to 100% of the dye was detected in the dissolution medium. This outcome was expected, as the ColoPulse coating can only dissolve at the pH peak of 7.4 in the simulated terminal ileum phase but not at pH 6.0 in the simulated colonic phase. As the simulated terminal ileum phase lasts for 30 min, the thicker coating layers of 10 and 12 mg/cm^2^ did not dissolve completely in that short period, as observed during the release testing. As a consequence, the remaining coating formed a barrier that reduced the drug release rate.

ColoPulse-coated tablets that underwent ethanol vapor treatment also started the release of methylene blue in GISS phase III; however, at a much lower rate across the three coating thicknesses. This can be explained by the fact that the ColoPulse layer of these tablets only opened on the tablet’s side in GISS phase III during the release experiment, leaving the top and bottom partially intact.

For enteric coating, a thickness of 8 mg/cm^2^ was applied to the printed tablets. This coating thickness resulted in a desirable release profile, regardless of whether or not the tablets were treated with ethanol vapor (Figure 7). Specifically, no release was observed in the first 2 h (the gastric phase) of the GISS. However, when the dissolution medium was switched to GISS phase II, simulating the jejunum, methylene blue was gradually released from the tablet, and after approximately 5 to 6 h, 95 to 110% methylene blue was released from both types of tablets.

To further confirm that the coating could effectively cover the entire surface of the PBP tablets, SEM imaging that zoomed in on the tablet surface was conducted. Tablets without ethanol vapor treatment coated with either ColoPulse or enteric coating were evaluated, as they originally had a rougher surface and thus might be more difficult to cover properly. In Figure 8, it can be observed that although there were small holes on the tablet surface, the coating material could penetrate and cover these areas when the magnification level was increased. This aligned with the dissolution data showing a desirable release profile for each coating solution.

The sustained release from the coated tablets after the coating opened (phase II or III in the GISS) can be attributed to both the high binder content of tablet E, consistent with findings from other studies, and the increased jetting liquid exposure at a lower layer height, leading to increased binding [10,30,31]. This sustained release might be beneficial in some cases when a more homogeneous spread of incorporated biopharmaceuticals along specific parts of the intestinal tract is needed. If a faster release is required, some modifications in the formulation, such as adding super disintegrants such as sodium starch glycolate or croscarmellose sodium, can be considered. These two disintegrants were found to drastically improve the dissolution of captopril in another 3DP product, even when 65% of the formulation contained a high molecular weight polymer binder, a binder content even higher than that used in this study [29].

Overall, the ability to directly apply different coatings on PBP tablets was a drastic improvement over the previously used sub-coating method, decreasing both the complexity and labor intensity. Previously, only 3DP tablets produced by fuse-deposition modeling (FDM) were studied for coating application, most likely due to their relatively smooth surfaces, which might have a positive effect on their coatability [10]. Nevertheless, the high process temperature of FDM hinders its application for biopharmaceuticals. Other efforts investigating controlled-release 3DP products primarily focused on printing an extra layer covering the core tablet to exhibit a desired dissolution profile [32,33,34,35,36,37,38]. These approaches may require efforts to develop the formulation and printing of those layers. This is circumvented by the work in this study, enabling the use of conventional coating techniques to directly coat PBP tablets with a controlled release layer without the need for further development. Importantly, the manipulation of formulation and printing parameters is straightforward to implement and does not affect the complexity of the overall production process, particularly given that printer parameters are software-controlled. In cases where adjustment of the printing process would not have a sufficient smoothening effect, possibly when using more viscous coating formulations, the ethanol vapor treatment might be added as an additional smoothening method. The changes observed in product characteristics due to the optimization of PBP tablets for direct coating application might also be useful for other purposes. Increasing binder content might be a possible option when a sustained release is needed. For PBP tablets requiring additional mechanical strength, a short ethanol vapor exposure might be sufficient. Furthermore, changing the layer thickness during printing can also lead to an increase in disintegration time. These modifications might be utilized when altering the printing powder formulation is not an option.

### 3.3. The Stability of Incorporated BIAP in PBP Tablets

The stability of encapsulated biopharmaceuticals in coated PBP tablets is of paramount importance. In this study, the model protein BIAP was subjected to several processes that may affect the integrity of the protein, i.e., spray drying, PBP, ethanol vapor exposure, coating, and storage. Tablets E were printed containing spray-dried inulin/BIAP with a ratio of 99:1 (*w*/*w*), and the binder:bulk:SD powder ratio was 50:45:5 (*w*/*w*/*w*). The enzymatic activity of BIAP was used as a readout to assess protein stability during each of these processing steps. The results are given in Figure 9.

Spray-drying did not affect BIAP activity; activity assays showed approximately 104% of enzymatic function remained after the process (Figure 9). No significant difference was found in the enzymatic activity between the spray-dried powder and printed tablet E, with or without ethanol vapor treatment (60 min) affecting enzymatic activity, even after 2 months of storage at 2–8 °C. After coating the tablets with either the ColoPulse or enteric coating, no significant change in enzymatic activity was observed compared to the spray-dried inulin:BIAP powder. One exception was tablet E pretreated with ethanol vapor after the enteric coating process showed reduced enzymatic activity. However, this might be due to the abnormal dissolution of one sample, as it was observed that, for unknown reasons, it did not dissolve completely. Overall, even for ethanol-treated tablet E, which had the highest level of solvent exposure (lowest layer thickness at the same line spacing), BIAP activity was maintained after all the steps. Therefore, it can be concluded that, for BIAP to exhibit its activity in the colon or intestine properly, the tablet modification methods used in this study did not result in instability of the incorporated protein. Additionally, the observed sustained drug release of these tablets might be beneficial for the use of BIAP in the treatment of ulcerative colitis, as the enzyme is distributed more equally across the treatment site, i.e., the colon. The feasibility of the direct coatability of PBP-printed products without affecting API stability opens up possibilities for further exploration of PBP dosage forms.

## 4. Conclusions

To the best of our knowledge, this is the first study to demonstrate that by manipulating the formulation and the printing process of PBP tablets, a relatively smooth surface can be produced that can be directly supplied with a ColoPulse or enteric coating without requiring a sub-coating. Directly coatable tablets could be produced by a proper combination of binder content, line spacing, and layer thickness. Alternatively, we demonstrated that tablets can be treated with ethanol vapor to further improve surface smoothness, with a concurrent increase in mechanical strength. The encapsulated BIAP in the modified tablets exhibited excellent stability after 2 months of storage as well as after the coating procedure.

## Figures and Tables

**Figure 1 pharmaceutics-15-02193-f001:**
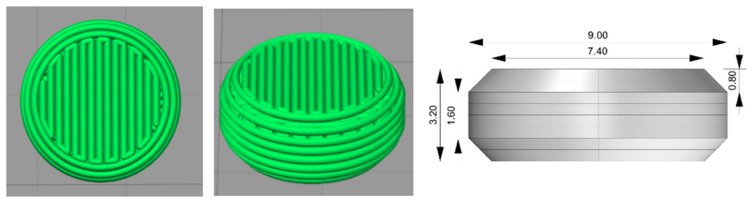
Deposition pattern of jetting liquid and tablet dimension.

**Figure 2 pharmaceutics-15-02193-f002:**
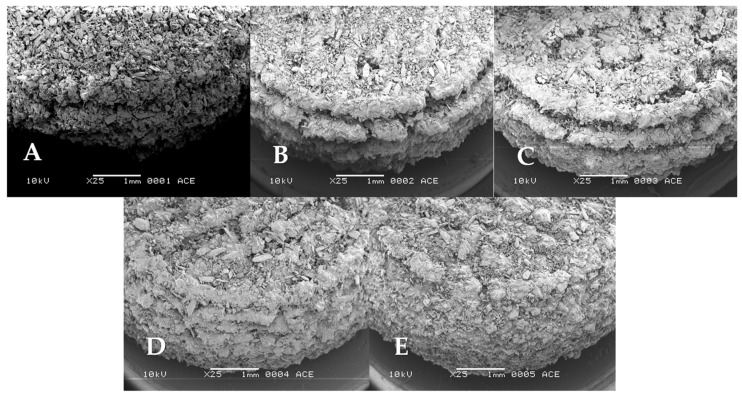
SEM micrographs of tablets (**A**–**E**).

**Figure 3 pharmaceutics-15-02193-f003:**
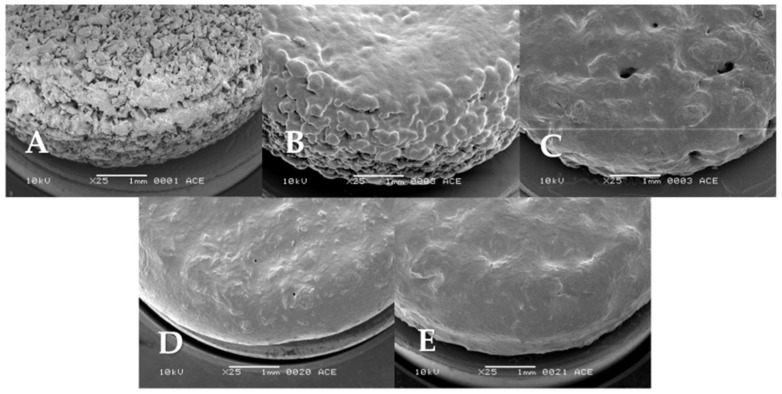
SEM micrographs of tablets (**A**–**E**) after 100 min of ethanol vapor treatment.

**Figure 4 pharmaceutics-15-02193-f004:**
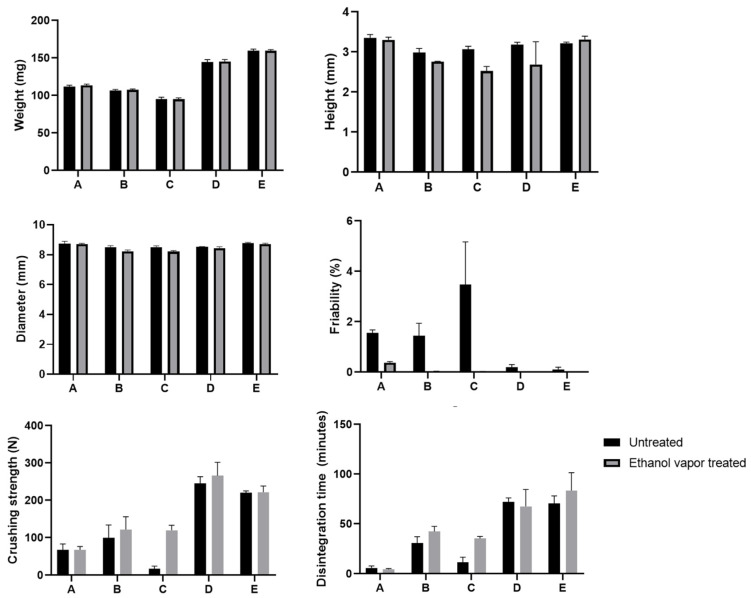
Physical properties of PBP tablets for different tablets, with and without ethanol vapor treatment for 100 min (*n* = 3).

**Figure 5 pharmaceutics-15-02193-f005:**
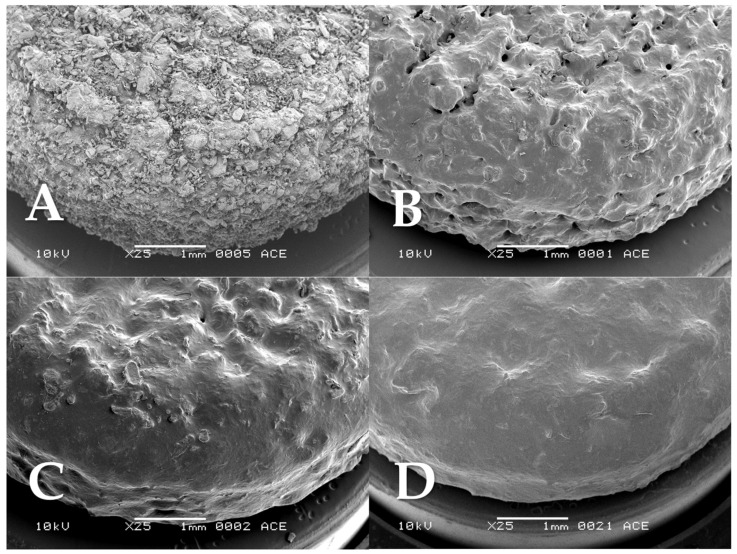
SEM micrographs of tablets E being treated with ethanol with different durations ((**A**): no treatment, (**B**): 15 min, (**C**): 30 min, and (**D**): 60 min).

**Figure 6 pharmaceutics-15-02193-f006:**
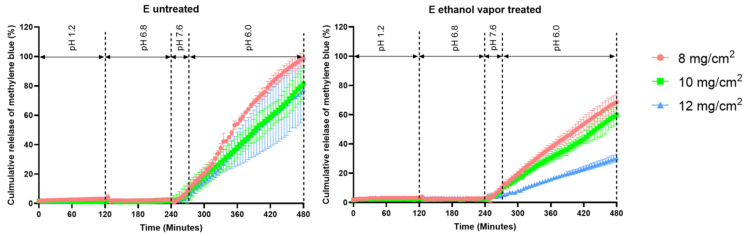
Release profile of methylene blue from PBP tablets E coated with ColoPulse at 8, 10, and 12 mg/cm^2^ thickness (*n* = 3).

**Figure 7 pharmaceutics-15-02193-f007:**
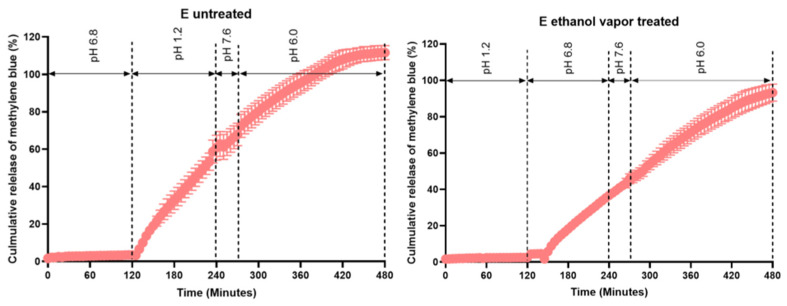
Release profile of methylene blue from PBP tablets E coated with ColoPulse at 8 mg/cm^2^ thickness (*n* = 3).

**Figure 8 pharmaceutics-15-02193-f008:**
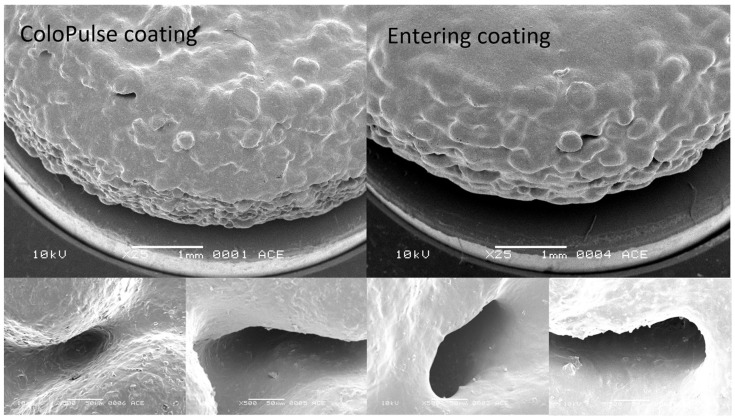
SEM micrographs of tablet E coated with ColoPulse and enteric coating 8 mg/cm^2^ were taken at 25× magnification on the top and zoomed in at 500× magnification on the bottom.

**Figure 9 pharmaceutics-15-02193-f009:**
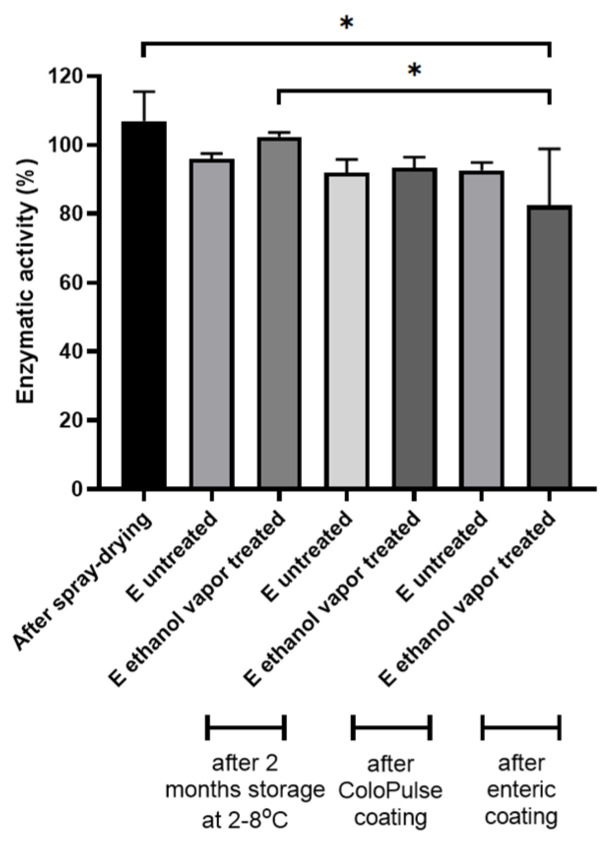
Enzymatic activity of alkaline phosphatase in tablets E (with and without ethanol vapor treatment) following storage for uncoated tablets and after different coating procedures (*n* = 3, * *p* < 0.05).

**Table 1 pharmaceutics-15-02193-t001:** Specifications of the GISS according to reference [23], adapted with permission [27].

Phase	Segment Gastrointestinal Tract	pH	Volume (mL)	Time (h)
I	Stomach	1.20 ± 0.20	500	2.0
II	Jejunum	6.80 ± 0.20	629	2.0
III	Terminal ileum	7.63 ± 0.12	940	0.5
IV	Colon	6.00 ± 0.25	1000	3.5

**Table 2 pharmaceutics-15-02193-t002:** Composition of the switch solutions in this study, adapted with permission [27].

Phase	Composition	Time Added to the Dissolution Vessel (h)
I	0.50 g sodium chloride, 1.75 mL concentrated hydrochloric acid, add demineralized water to 250 mL	0
I to II	2.04 potassium dihydrogen phosphate, 15 mL sodium hydroxide 2.0 M (80 g/L), add demineralized water to 65 mL	2.0
II to III	1.02 g potassium dihydrogen phosphate, 6.0 mL sodium hydroxide 2.0 M (80 g/L), add demineralized water to 156 mL	4.0
III to IV	4.5 mL hydrochloric acid 3.0 M, add demineralized water of 30 mL	4.5

**Table 3 pharmaceutics-15-02193-t003:** Formulations and printing conditions.

Tablet	Binder: Bulk Ratio (*w*/*w*)	Layer Height (mm)	Line Spacing (mm)
A	20:80	0.4	0.45
B	50:50	0.4	0.45
C	50:50	0.4	0.50
D	50:50	0.2	0.50
E	50:50	0.1	0.50

## Data Availability

Not applicable.

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
