# Peer review of "Surface Engineering Methods for Powder Bed Printed Tablets to Optimize External Smoothness and Facilitate the Application of Different Coatings"

_pharmaceutics, 2023, doi:10.3390/pharmaceutics15092193_

Round 1

Reviewer 1 Report

Dear Authors,

your work is thorough and very detailed. I feel that the subject has fallen victim to over-technologisation.

In the introductory section, I miss the presentation of the concrete advantiges of 3d printing. Please expand the manuscript by explaining it.
I miss the justification and explanation of why powder-bed printing method is better than other formulation methods.

line 39-41 :

"This versatility, together with the flexibility to manufacture in small volumes, greatly benefits medical practice, drug development as well as clinical testing of newly developed drugs that exist in limited quantities [1]."

This statement is not adequately supported - Versatility, together with the flexibility to manufacture in small quantities, is a suggested reference to support the statement:

https://doi.org/10.3390/pharmaceutics13101571

https://doi.org/10.1208/s12249-021-02066-y

https://doi.org/10.1208/s12249-019-1500-2

Fig2, 3 and 5 sizes are small. Please use the Fig8 size adjustment.

The description of the experiments is very detailed.

Question 1: If more binder had been used for printing, would the surface roughness of the tablet have been reduced?

Question 2: If a larger amount of binder was used to print the tablet, would a heat treatment at a given temperature not be sufficient instead of ethanol treatment?

The heat treatment softens the tablet surface and reduces the surface roughness.

It would be worthwhile to carry out some new experiments using larger amounts of binder combine with heat treatment at a given temperature.

Fig2, 3 and 5 are small. Please use the Fig 8 size setting.

Author Response

Dear reviewer,

Please find our response to your comment attached as a word document.

Kind regards,

Wouter Hinrichs

Reviewer 2 Report

The manuscript 'Surface engineering methods for powder bed printed tablets to optimize external smoothness and facilitate the application of different coatings' deserves attention. The study demonstrated the successful production of PBP tablets with surfaces suitable for direct tablet coating with preserved stability of the encapsulated BIAP. The technologies used, the applied analytical methods, and the obtained results are described in detail and supplemented with evidentiary material - figures and microphotographs. Furthermore, the results are thoroughly discussed. I have no critical remarks about the authors and recommend to the respected editors that the manuscript be accepted for publication in its current form.

Author Response

Dear reviewer,

We are grateful for the kind words and are pleased that you found the manuscript of sufficient quality!

Kind regards,

Also on behalf of my co-authors,

Wouter Hinrichs

Reviewer 3 Report

The authors detailed a novel method to optimize external smoothness of powder bed printed tablets. The authors also claim that the new method can be used to apply coating without directly sub-coating. However, treating tablets using ethanol also seems to be an additional processing step with its own complexity like endpoint determination, temperature effect etc. Please justify.

Author Response

Dear reviewer,

Please find attached our response to your comments.

Best regards,

Daan Zillen

Round 2

Reviewer 1 Report

Dear Authors,

please add the following reference to the manuscript (line 37)

Haimhoffer, Á.; Fenyvesi, F.; Lekli, I.; Béresová, M.; Bak, I.; Czagány, M.; Vasvári, G.; Bácskay, I.; Tóth, J.; Budai, I. Preparation of Acyclovir-Containing Solid Foam by Ultrasonic Batch Technology. Pharmaceutics 202113, 1571. https://doi.org/10.3390/pharmaceutics13101571

Author Response

Dear reviewer,

We have added the reference to line 37 and are glad you found the manuscript otherwise of sufficient quality to be published.

Kind regards,

Wouter Hinrichs

Reviewer 3 Report

The authors have adequately responded to my comment.

Author Response

Dear reviewer,

Thanks for the feedback!

Kind regards,

Wouter Hinrichs